# Crystal Engineering of Ionic Cocrystals Sustained by Azolium···Azole Heterosynthons

**DOI:** 10.3390/pharmaceutics14112321

**Published:** 2022-10-28

**Authors:** Maryam Rahmani, Vijith Kumar, Julia Bruno-Colmenarez, Michael J. Zaworotko

**Affiliations:** Department of Chemical Sciences and Bernal Institute, University of Limerick, V94 T9PX Limerick, Ireland

**Keywords:** crystal engineering, ionic cocrystals, charge-assisted hydrogen bond, azolium···azole, supramolecular heterosynthon

## Abstract

Crystal engineering of multi-component molecular crystals, cocrystals, is a subject of growing interest, thanks in part to the potential utility of pharmaceutical cocrystals as drug substances with improved properties. Whereas molecular cocrystals (MCCs) are quite well studied from a design perspective, ionic cocrystals (ICCs) remain relatively underexplored despite there being several recently FDA-approved drug products based upon ICCs. Successful cocrystal design strategies typically depend on strong and directional noncovalent interactions between coformers, as exemplified by hydrogen bonds. Understanding of the hierarchy of such interactions is key to successful outcomes in cocrystal design. We herein address the crystal engineering of ICCs comprising azole functional groups, particularly imidazoles and triazoles, which are commonly encountered in biologically active molecules. Specifically, azoles were studied for their propensity to serve as coformers with strong organic (trifluoroacetic acid and p-toluenesulfonic acid) and inorganic (hydrochloric acid, hydrobromic acid and nitric acid) acids to gain insight into the hierarchy of NH^+^···N (azolium-azole) supramolecular heterosynthons. Accordingly, we combined data mining of the Cambridge Structural Database (CSD) with the structural characterization of 16 new ICCs (11 imidazoles, 4 triazoles, one imidazole-triazole). Analysis of the new ICCs and 66 relevant hits archived in the CSD revealed that supramolecular synthons between identical azole rings (A^+^B^−^A) are much more commonly encountered, 71, than supramolecular synthons between different azole rings (A^+^B^−^C), 11. The average NH^+^···N distance found in the new ICCs reported herein is 2.697(3) Å and binding energy calculations suggested that hydrogen bond strengths range from 31–46 kJ mol^−1^. The azolium-triazole ICC (A^+^B^−^C) was obtained via mechanochemistry and differed from the other ICCs studied as there was no NH^+^···N hydrogen bonding. That the CNC angles in imidazoles and 1,2,4-triazoles are sensitive to protonation, the cationic forms having larger (approximately 4.4 degrees) values than comparable neutral rings, was used as a parameter to distinguish between protonated and neutral azole rings. Our results indicate that ICCs based upon azolium-azole supramolecular heterosynthons are viable targets, which has implications for the development of new azole drug substances with improved properties.

## 1. Introduction

Crystal engineering [1] is the field of chemistry that studies the design, properties, and applications of crystalline solids [2,3] and offers promise for the design of new materials with improved properties for a number of applications including porous materials gas separation and storage [4,5,6], polar crystals for non-linear optics [7,8], and the use of pharmaceutical cocrystals as improved drug substances [9,10,11,12]. With respect to pharmaceutical science, crystal engineering can also offer insight into other classes of crystalline drug substances including their propensity to form polymorphs and multi-component molecular crystals such as solvates, hydrates, and cocrystals [13,14]. Cocrystals, defined as “solids that are crystalline single phase materials composed of two or more different molecular and/or ionic compounds generally in a stoichiometric ratio which are neither solvates nor simple salts” [13], are of relevance to pharmaceutical science because they can modify the physicochemical properties of a drug molecule without changing its chemical structure. Cocrystals can thereby offer improved, sometimes greatly improved solubility [15,16], stability [17], dissolution rate [18], and bioavailability [19,20]. It is therefore unsurprising that the increased interest in the study of pharmaceutical cocrystals that started in the early 2000s [21,22,23,24] has resulted introduction of new drug products based upon pharmaceutical cocrystals [11,25].

Cocrystals can be classified as molecular cocrystals (MCCs) or ionic cocrystals (ICCs). MCCs are crystalline solids comprised of at least two neutral molecular compounds (coformers) sustained by noncovalent bonds, e.g., hydrogen [3] or halogen bonds [26], in a stoichiometric ratio. ICCs are comprised of at least one salt and a coformer [27] and typically sustained by charge-assisted supramolecular synthons or, if metals are present, coordination bonds (therefore ICCs and coordination networks may not be mutually exclusive) [17,28]. Indeed, the first cocrystal was an ICC of NaCl and urea reported 1783 by De I’Ise, who noted a change in morphology of NaCl crystals obtained from slow evaporation urea-containing solutions (e.g., urine) [29]. Subsequently, ICCs of NaCl and sugars were reported in the literature [30,31,32,33]. In the 1960s, the crystal structure of sodium chloride and urea was reported by Palm and MacGillivray, after Kleber detailed the morphology and optics of this compound [34]. In 2004, Childs et al. presented an early example of a pharmaceutical ICC that used carboxylic acid coformers to regulate the dissolution rate of fluoxetine hydrochloride (Prozac^®^) [23]. To describe the sodium bromide-barbituric acid cocrystal, Braga coined the term “ionic cocrystal” in 2010 [27]. This nomenclature however is not universal, with terms such as “complex salts”, “metal salt complexes”, “acid salts”, “salt-cocrystals”, “adducts”, and “hybrid salt-cocrystals” also being used for ionic cocrystals [35,36,37,38,39]. Most recently, systematic studies of ICCs have been conducted by our group through the study of phenol-phenolate ICCs sustained by PhOH···PhO^−^ supramolecular heterosynthons [40] and phosphoric acid-dihydrogen phosphate ICCs [41].

ICCs are almost always sustained by charge-assisted hydrogen bonding, which can be advantageous as they are typically shorter and stronger than hydrogen bonds formed between two neutral groups [42]. That they are necessarily comprised of three or more components means that ICCs also offer greater compositional diversity than MCCs, e.g., A^+^B^−^C, where A^+^ = cation, B^−^ = anion and C = neutral coformer. This contrasts with MCCs, most of which are 2-component AB cocrystals. There is also the possibility of ICCs in which a free base serves as a coformer with a salt of that free base or the free acid serves as a coformer with a salt of that acid, i.e., A^+^B^−^A or A^+^B^−^B, respectively. Such ICCs are attractive in pharmaceutical science since there would be a relatively high mass % of the drug compound, reducing the drug dosage. An FDA-approved cocrystal that comprises an A^+^B^−^B drug substance is Depakote^®^. In addition, there are pharmaceutical ICCs involving two different drug molecules that serve as coformers. Entresto^®^ (a hydrated A^+^B^−^B′-co-crystal comprised of the sodium salts of sacubitril and valsartan) [43] and Seglentis^®^ (an A^+^B^−^C co-crystal where C = celecoxib, an anti-inflammatory, A^+^ = protonated tramadol, an analgesic, and B = chloride) [25,44]. Nevertheless, from a crystal engineering perspective, ICCs remain understudied as shown by an investigation into multicomponent crystal structures archived in the CSD by the Grothe group, which revealed that the number of such ICCs (2.1%) was much lower than MCCs (10.6%) [35]. As such, it is plausible to assert that ICCs are a long-known but understudied class of multicomponent crystals.

The design of cocrystals involves understanding of the hierarchy of supramolecular synthons as the strongest hydrogen bond donors-acceptors would be expected to dominate. This is one of Etter’s rules [45,46]. In this regard, charge-assisted hydrogen bonds can drive the formation of supramolecular heterosynthons, including between chemical species that cocrystallize as conjugate acid-bases [47]. Whereas dicarboxylate salts, [COO-HOOC]^−^, are quite well studied [39], [N-H···N]^+^ hydrogen bonds are underexplored from a crystal engineering perspective, with pyridines being the most studied thus far [48,49]. The relative paucity of systematic crystal engineering studies is despite azole compounds being appealing targets due to their presence in biologically active compounds [50]. Our survey of the DrugBank database (v 5.1.7) [51] revealed that 12.8% (349/2721) of approved small molecule drug substances contain at least one azole moiety. There are three main types of azoles, N-, N/O-, and N/S-containing azoles, with N-azoles accounting for 246 drug substances (almost 70% of azole-based drug molecules). Imidazole and triazole derivatives offer a broad range of biological activity, including anti-fungal, anti-inflammatory, anti-platelet, anti-microbial, anti-mycobacterial, anti-tumoral, and antiviral properties [52,53,54,55,56], and form hydrogen bonds with drugs and proteins [57,58]. However, due to their poor aqueous solubility (many are BCS class II [59]), the bioavailability of this family of medicines can be a challenge. In this context, developing new salt forms of basic drug compounds has traditionally been the approach taken [60,61], although increasingly MCCs are being studied to improve key physicochemical properties such as solubility, bioavailability and stability [62,63,64,65,66]. ICCs comprising azolium compounds are of interest not just because their proven biological activity [67,68], they are also relevant to molecular sensors [69,70,71], catalysis [72,73], and energetic materials [74,75]. Indeed, to date most azolium···azole ICCs were studied in the context of energetic materials through ionic liquids and salts comprised of nitrogen-rich molecules [76,77]. Herein, we report a systematic crystal engineering study that evaluates the potential of azole compounds to reliably form ICCs.

## 2. Materials and Methods

All reagents and solvents were obtained commercially and utilized without additional purification.

### 2.1. Powder X-ray Diffraction (PXRD)

PXRD patterns were obtained using a PANalytical Empyrean™ diffractometer equipped with a PIXcel3D detector with the following experimental parameters: Cu Kα radiation (λ = 1.54056 Å); 40 kV and 40 mA; scan speed 8°/min; step size 0.05°, 2θ = 5–40°.

### 2.2. Single-Crystal X-ray Diffraction

SCXRD data for compounds 1–6, 10–12, 14 and 15 were collected using a Bruker Quest D8 Mo Sealed Tube equipped with CMOS camera and Oxford cryosystem with MoKα radiation (wavelength of λ = 0.7103 Å). SCXRD data for compounds 7, 8, 13, 16 and 17 were collected on a Bruker Quest D8 Cu Microfocus with CuKα radiation (wavelength of λ = 1.5418 Å). X-ray measurements were made using APEX 4 software, frames were integrated with Bruker SAINT [78] software and absorption corrections were performed using multi-scan methods. Crystal structures were solved by direct methods using OLEX2 [79] and anisotropic displacement parameters for non-hydrogen atoms were applied. Some hydrogen atoms were placed at calculated positions and treated using a riding model whereas other H-atoms were located in the Fourier difference maps and placed geometrically.

### 2.3. ICC Design (Coformer Selection)

Several criteria were applied for selecting coformers (Figure 1): (i) imidazole and 1,2,4-triazole moieties should have sp^2^ N acceptor atoms that are sterically accessible; (ii) only one azole group should be present to preclude intramolecular hydrogen bonding interactions after protonation; (iii) coformers should be free of competing protonation sites; (iv) organic and inorganic acid coformers (Figure 1) must be acidic enough to prevent molecular cocrystal formation (ΔpK_a_ > 3.7) and contain just one acidic hydrogen. The ΔpK_a_ rule [80] was taken into account as the azole-based molecules and acids studied herein have ΔpK_a_ values from 6.17 to 21.06 (ΔpK_a_ ≥ 3.7), consistent with proton transfer with the selected acids (Appendix A).

### 2.4. Synthesis of Ionic Cocrystals

To synthesize azolium-azole ICCs, two approaches were followed (Figure 2). Details are presented in the Appendix A.

#### 2.4.1. Approach I (Solvent Evaporation)

Approach I relied on slow evaporation of solutions of each azole derivative (A or C) in a 2:1 stoichiometric molar ratio with an acid. In this study, several solvent systems were used for ICC screening (water, methanol, ethanol, and acetonitrile, or a combination of these solvents). Suitable single crystals for single crystal X-ray diffraction were isolated and bulk samples were tested using X-ray powder diffraction (see Table 1 and Appendix A for crystallographic details).

#### 2.4.2. Approach II (Solvent-Drop Grinding)

ICCs were prepared using A^+^B^−^ salts with a 1:1 ratio of azole (A) and acid (B) (Appendix A) that were subjected to solvent-drop grinding (SDG) by adding 1 molar ratio of azole as a coformer (A) to generate type A^+^B^−^A cocrystals (comprising AH^+^···A supramolecular heterosynthons) or a different neutral coformer (C) to generate A^+^B^−^C ICCs (comprising AH^+^···C supramolecular heterosynthons). PXRD was used to characterise the microcrystalline products (see Appendix A for comparison of calculated and experimental PXRD patterns, which are consistent with each other).

### 2.5. Computational Methods

The intermolecular interaction energies of charge-assisted azolium-azole hydrogen bonds were calculated using monomer wavefunctions at the B3LYP/6-31G(d,p) level in the CrystalExplorer 17.5 [81] program package followed by geometry optimization carried out using the CASTEP module with GGA-type PBE functional contained in Materials Studio 8.0. The total interaction energy was divided into electronic (E_ele_), polarization (E_pol_), dispersion (E_dis_), and repulsion (E_rel_) components.

### 2.6. Cambridge Structural Database (CSD) Analysis

A CSD survey was conducted using ConQuest (v.5.43, November 2021) and the results were processed with Mercury (v.2021.3.0). Restrictions are detailed in Section 3 of the SI. ICC entries were manually filtered from the resultant hitlist (Appendix A). The findings of the CSD search for azolium-azole heterosynthons are presented in [Fig pharmaceutics-14-02321-ch001] and [Fig pharmaceutics-14-02321-ch002]. The following parameters were evaluated: (i) azole ring arrangements in azolium-azole ICCs; (ii) charge assisted NH^+^···N heterosynthons between identical (AH^+^···A) or different (AH^+^···C) azoles; (iii) average distances and angles for NH^+^···N hydrogen bonds; (iv) average CNC angles within neutral and protonated imidazole or 1,2,4-triazole rings.

## 3. Results and Discussion

### 3.1. CSD Analysis of Azolium-Azole Supramolecular Heterosynthons

The CSD survey revealed 211 crystal structures of azole moieties that involve azolium···azole supramolecular heterosynthons, 132 (63%) salts, 66 ICCs (31%) and 13 zwitterions (6%) (Appendix A). 9-ethylguanine hemi-hydrochloric acid, the earliest azolium-azole ICC entry in the CSD, was reported in 1975 and exhibited the shortest NH···N hydrogen bond (2.637 Å) observed at the time [82]. When two or more azole groups are present in the same crystal structure, two types of NH^+^···N supramolecular heterosynthons are possible, those involving the same azole, AH^+^···A (56) or those involving different azoles, AH^+^···C (10). Our analysis of the 40 imidazole-containing hits in the 66 archived ICCs (Appendix A and Appendix A) revealed significantly more AH^+^···A (38) than AH^+^···C (2) supramolecular heterosynthons. Moreover, more AH^+^···A (10) than AH^+^···C (2) supramolecular heterosynthons were in the 12 hits involving 1,2,4-triazole rings. Despite there being hits for other azolium···azole interactions, as indicated in Appendix A, their small quantity (e.g., 9 hits for pyrazolium-pyrazole and 1 hit for 1,2,3-triazolium-1,2,3-triazole) precludes statistical evaluation.

That 79% of previously reported azolium···azole ICCs are sustained by NH···N supramolecular heterosynthons involving imidazoles or 1,2,4-triazoles suggests that charge-assisted NH^+^···N interactions could be generally suitable for ICC formation via the motif in Figure 3; this is the primary focus of this contribution.

### 3.2. Structural Parameters That Distinguish between Azole and Azolium Rings

The location of hydrogen atom positions can be disputed due to the low electron density of hydrogen atoms which means that determination of their coordinates by X-ray diffraction experiments is challenging. Fortunately, the hydrogen atom position in an NH^+^···N supramolecular heterosynthon may be verified using the azole ring structural parameters, as shown by Rogers’s and co-workers in their studies on imidazolium salts [83]. Further, our prior work on pyridines examined the use of the CNC angle in aromatic rings to distinguish between protonated and free base pyridines [84].

To address this issue, we identified key parameters in crystal structures of neutral and protonated imidazoles or 1,2,4-triazoles (Figure 4) to distinguish between azole and azolium moieties.

Histograms of hydrogen bond parameters in azolium-azole ICCs and CNC angles in neutral and cationic (θ_1_ and θ_2_) imidazoles and 1,2,4-triazoles are given in [Fig pharmaceutics-14-02321-ch001] and [Fig pharmaceutics-14-02321-ch002], respectively. For 11360 neutral imidazoles (θ_1_), the average CNC angle was determined to be 104.9(10)°. In comparison, 1868 cationic imidazoles revealed a CNC angle (θ_2_) averaging 109.2(8)°. The corresponding values were 102.3(9)° and 106.7(6)° for 1,2,4-triazole (2414 hits) and 1,2,4-triazolium (72 hits) rings, respectively. This difference in average θ_1_ and θ_2_ values (approximately 4 degrees) in imidazoles and 1,2,4-triazoles clearly illustrates that they are sensitive to protonation, and their cationic forms have higher values (around 4.4 degrees) than comparable neutral rings, allowing us to distinguish between protonated and neutral rings.

### 3.3. Crystal Structure Descriptions

SE and SDG approaches were applied to prepare seventeen ICCs (ICC 2 is previously reported in the CSD as BEPTEX [84]), 16 of which are A^+^B^−^A ICCs sustained by NH^+^···N supramolecular heterosynthons. SDG (ICCs 1, 3–9, 11, 12, 14–16) and SE (ICCs 2, 4–16) approaches yielded similarly high success rates with respect to isolation of A^+^B^−^A ICCs (Appendix A). Several cocrystals that did not form by SE were obtained by SDG in the same solvent, or vice versa. Table 1 lists the crystallographic parameters of all 17 ICCs obtained in this study and Figure 1 presents their ORTEP diagrams.

#### 3.3.1. ICCs Containing 1,2-Dimethylimidazole, DMI (1 and 2)

SCXRD revealed that the two ICCs obtained with DMI (**DMIHBA** and **BEPTEX**) are isostructural and that both crystallize in the orthorhombic space group *Pca*2_1_ (Appendix A). As noted in Table 1, the crystal structure of **BEPTEX** was previously reported [84], however in order to do a detailed analysis on NH^+^···N hydrogen bonds, we include it in this report. **DMIHBA** and **BEPTEX** contain one free DMI molecule, one protonated DMI^+^ cation and one bromide or chloride anion, respectively. Charge-assisted NH^+^···N supramolecular heterosynthons (D_N_^…^_N_, 2.712(3) in **DMIHBA**, and 2.699(4) Å in **BEPTEX**) are present in both ICCs. The CNC angles of the imidazolium and imidazole rings (Table 2) are 108.54(24)° and 106.02(23)° for **DMIHBA** and 108.83(37)° and 106.17(36)° for **BEPTEX**, enabling identification of which ring is protonated.

#### 3.3.2. ICCs Containing 1-(4-Methoxyphenyl)-1H-Imidazole, MPI, 3–7

Cocrystallization of MPI was successful in all acids except for HCl, affording five new ICCs. HCl produced a physical mixture of MPI and related salt (Appendix A). SCXRD revealed that the MPI cocrystals each formed monoclinic crystals. **MPIHBA·2H_2_O** and **MPITFA** crystallized in *P*2_1_/*c*, **MPINIA·2H_2_O** and **MPITSA** crystallized in *P*2_1_/*n*, while **MPINIA** crystallized in space group *C*2/*c*. Their NH^+^···N supramolecular synthons and crystal packing are illustrated in Figure 3. The proton positions in **MPIHBA·2H_2_O** and **MPINIA** were found to be disordered between two imidazole rings, making it difficult to differentiate imidazole from imidazolium. While in other ICCs the θ_1_ and θ_2_ angles in MPI molecules support the proposed ionic nature of one of the MPI molecules, for **MPIHBA·2H_2_O** and **MPINIA** this was not the case.

In **MPIHBA·2H_2_O**, the D_N_^…^_N_ distances, 2.644(3) and 2.662(4) Å, are at the short end of the range of distances for this type of interaction. In **MPIHBA·2H_2_O**, an isolated site hydrate ICC, water molecules interact with bromide anions through $R_{4}^{2}\left(8\right)$ and $R_{6}^{4}\left(12\right)\;$ graph set motifs (Appendix A). Two ICCs of MPI with nitric acid were isolated. **MPINIA** is an anhydrate with a disordered hydrogen atom in NH^+^···N hydrogen bond, whereas **MPINIA·2H_2_O** is an isolated site hydrate. The D_N_^…^_N_ in **MPIHCA** is 2.652(1) Å, which is shorter than the 2.678(1) Å length in **MPINIA·2H_2_O**. In the crystal structure of **MPINIA·2H_2_O**, the values of θ_1_ and θ_2_ in imidazole rings are 105.79(10)° and 108.67(11)°, respectively, allowing identification of the protonated ring, whereas in **MPINIA** there is no statistical difference between the two rings (Table 2). Water molecules in **MPINIA·2H_2_O** form chains with nitrate anions ($C_{3}^{3}\left(8\right)$ motifs) along the crystallographic b-axis (Appendix A). **MPITSA** and **MPITFA** are comprised of one MPI free base, one protonated MPI^+^ and one anion (p-TSO^−^/OTf^−^), which are both sustained by NH^+^···N hydrogen bonds with distances of 2.791(4) and 2.748(3) Å, respectively. In **MPITSA**, the values of θ_1_ and θ_2_ in the imidazole rings are 104.90(31)° and 109.00(26)°, respectively, and 105.44(22)° and 108.70(23)° in **MPITFA**. The orientation of the imidazole ring in relation to the benzene moiety in **MPIHBA·2H_2_O**, **MPINIA**, **MPINIA·2H_2_O**, **MPITSA**, and **MPITFA** was determined to be 27.23(38)°, 44.38(24)°, 17.05(18)°, 17.58(43)°, and 24.50(21)°, respectively, suggesting that the MPI molecule in **MPINIA·2H_2_O** has the most planar structure in the MPI-based ICCs, while **MPINIA** has the least (Figure 2). In addition, the orientation of the imidazole ring to the methoxy group in **MPIHBA·2H_2_O** and **MPITFA** differs from that of others.

#### 3.3.3. ICC Containing 4-(1H-Imidazol-1-yl) Benzaldehyde, IMB, 8

Attempts to prepare IMB ICCs resulted in the formation of only one ICC (**IMBHBA**), with most attempts resulting in a physical mixture of starting materials or new solid forms with poor-quality crystals. SCXRD revealed that **IMBHBA** crystallizes in a monoclinic space group *C*2/*c* (Table 1), and the asymmetric unit comprises one IMB cation, one free IMB molecule and well as one-half of the anion (Br^−^) and one IMB cation with a disordered hydrogen atom (Figure 3). In IMBHBA, D_N_^…^_N_ distances are 2.678(4) and 2.658(4) Å, and the values of θ_1_ and θ_2_ are 107.29(30)° and 108.25(30)° in one pair of IMBs, and 107.05(31)° in paired IMBs sustained by disordered NH^+^···N hydrogen bond.

#### 3.3.4. ICCs Containing 1-(4-Nitrophenyl)-1H-Imidazole, NPI, 9–12

**NPIHBA** crystallized in the triclinic space group *P*$\bar{1}$, the asymmetric unit comprising an NPI cation, a free NPI and one bromide anion. D_N_^…^_N_ in **NPIHBA** is 2.686(2) Å and the CNC angles of the imidazolium and imidazole rings are 106.08(17)° and 109.23(17)°, respectively. SCXRD revealed that the **NPIHCA** asymmetric unit is comprised of one NPI cation, one free NPI molecule and one Cl^−^ anion. The value of D_N_^…^_N_ in **NPIHCA** is 2.691(2) Å, while the values of θ_1_ and θ_2_ are 106.46(1)° and 108.71(1)°, respectively. **NPINIA** crystallized in the monoclinic space group *C*2 and its asymmetric unit is similar to **MPINIA**, comprising one INP molecule with disordered hydrogen atoms and half a disordered nitrate anion (Figure 4). The D_N_^…^_N_ in **NPINIA** is 2.678(3) Å, shorter than the other ICCs that contain NPI molecules. As the hydrogen atom between the two imidazole rings is disordered, the imidazole angles θ_1_ and θ_2_ in this ICC are identical (107.83(23)°). When TFA was used as the acid, one free INP molecule, one INP cation and one triflate anion form the **NPITFA** asymmetric unit (Figure 4). In this structure, D_N_^…^_N_ is 2.808(1) Å (the longest D_N_^…^_N_ in this family of ICCs), and the 4 degree difference between θ_1_ and θ_2_ (105.30(12)° and 109.11(12)°, respectively) supports the ionic nature of one of the NPI molecules.

#### 3.3.5. ICCs Containing 1-(4-Methoxyphenyl)-1H-1,2,4-Triazole, MPT, 13–16

Four ICCs were formed by MPT, a 1,2,4-triazole. **MPTHBA** and **MPTHCA** crystallized in the triclinic space group *P*$\bar{1}$, their asymmetric units comprising one MPT^+^ cation, one MPT molecule and one bromide or chloride, respectively. In **MPTHBA** and **MPTHCA**, which are isostructural, NH^+^···N hydrogen bonds connect triazole and triazolium moieties at distances of 2.690(3) and 2.663(2) Å, respectively. **MPTHCA·2H_2_O** is an isolated site hydrate form of **MPTHCA** harvested from a 5:1 EtOH/H_2_O solution that crystallized in the triclinic space group *P*$\bar{1}$, with an asymmetric unit consisting of one MPT cation, one MPT molecule, one chloride anion, and two water molecules. Water molecules interact with chloride anions through $R_{4}^{2}\left(8\right)\;$ and $R_{6}^{4}\left(12\right)$ graph set-motifs (Appendix A). The D_N_^…^_N_ in **MPTHCA** and **MPTHCA·2H_2_O** are 2.663(2) and 2.723(3) Å. Moreover, the values of θ_1_ and θ_2_ are similar (103.44(16)° and 106.36(16)° in **MPTHCA**, 103.42(20)° and 106.04(20)° in **MPTHCA·2H_2_O**). **MPTTFA** crystallized in the monoclinic space group *C*2/*c* with one MPT^+^, one MPT and one triflate anion in the asymmetric unit (Figure 5). The D_N_^…^_N_ in this structure is 2.694(6) Å, and the values of θ_1_ and θ_2_ in triazole rings, 101.71(54)° and 105.70(52)°, allow identification of the protonated triazole.

#### 3.3.6. A^+^B^−^C Type ICCs

We also attempted to prepare azolium-azole ICCs comprised of different azoles (AH^+^···C). Such an approach could be useful to form ICCs containing azole-based APIs or pharmaceutical ICCs with azole coformers to improve physicochemical properties. Approaches I and II (Figure 2) resulted in a very low success rate, compatible with the conclusions of our CSD survey. Solvent evaporation with a 1:1:1 ratio of coformer 1, coformer 2 and acid generally resulted in a physical mixture of azole and the salt of the more basic azole. In terms of using a salt as a coformer, just one attempt by SDG was successful in forming azolium-azole ICC between two different azole compounds (**MPIBMPT**). **MPIBMPT** crystallized in the monoclinic space group *Cc* with one MPI^+^ cation, one chloride anion and one MPT molecule in the asymmetric unit. The absence of a charge-assisted NH···N^+^ supramolecular heterosynthon in **MPIBMPT** distinguishes it from the other ICCs reported herein (Figure 6). Rather, the MPI^+^ cation was found to have formed a hydrogen bond with the chloride anion (N2^+^···Cl1^−^ = 3.036(5) Å). MPI^+^ cations and MPT molecules were observed to form infinite chains of alternating cations and molecules, thereby enabling head-to-tail π···π stacking of azole and phenyl rings (Figure 6). The CNC angle in the imidazole ring is 108.52(54)°, which is within the range of protonated imidazoles, and the CNC angle in the triazole ring is 101.94(52)°, confirming that the triazole ring in this structure is neutral.

### 3.4. Overall Analysis of Crystal Structures

Several ICCs reported herein were found to exhibit disordered hydrogen atoms between azoles, so-called ‘confused protons’, a term coined in previous research on azole compounds [83,85]. In such structures, distinguishing between protonated and neutral azole rings is problematic but this does not affect the overall reliability of the approach of exploiting the NH^+^···N supramolecular heterosynthon for ICC formation. Figure 7 and Appendix A compare hydrogen bond parameters in neutral NH···N hydrogen bonds with those of charge-assisted interactions (NH^+^···N) from the CSD and the ICCs reported in this study. The interactions observed in this study, with distances ranging from 2.644(3) to 2.808(1) Å and angles approaching 180^°^ are consistently amongst the strongest based upon these structural parameters.

### 3.5. Hydrogen-Bond Strengths

The calculated energies of azolium azole hydrogen bonds at the B3LYP/6–31g (d,p) level of theory are shown in Table 3 for 13 ICCs (in which the hydrogen atoms’ positions in NH^+^···N supramolecular heterosynthons were not disordered). The contributions to total energies from electrostatic, polarization, dispersion, and repulsion interactions for NH^+^···N hydrogen bonds were also estimated. In terms of the azole rings involved in hydrogen bonds, the refined crystal structures of the ICCs described herein can consider two forms of hydrogen bonding, NH^+^···N supramolecular heterosynthons in imidazolium-imidazole ICCs and triazolium-triazole ICCs.

These results indicate binding energies 31.0 to 45.6 kJ mol^−1^, with **NPIHCA** showing the highest energy at 45.6 kJ mol^−1^. Electrostatic energy (E_ele_) has a stronger effect on E_tot_ than dispersion and polarization energies. Furthermore, the energies of imidazolium-imidazole and triazolium-triazole hydrogen bonding are comparable, as switching from imidazole in MPITFA (−41.4 kJ mol^−1^) to triazole in MPTTFA (−41.5 kJ mol^−1^) showed negligible impact. Conversely, substituting a methoxy group (**MPITFA**) for a nitro group in **NPITFA** (−44.6 kJ mol^−1^) had a more substantial impact. The azolium-azole dimer strength in ICCs containing the same coformers suggest that the impact of anions on binding energy is low (around 1–2 kJ mol^−1^), even when replacing inorganic anions with larger organic anions. The greater binding energies calculated for IMB/NPI-containing ICCs suggest that electronegative substituents may result in stronger NH···N hydrogen bonds. With hydrated ICCs in mind, we were curious to examine the energy of these interactions in the two ICCs that afforded both hydrate and anhydrate forms. The analysis in **MPINIA** failed due to disorder. By calculating the energy of NH^+^···N interactions in **MPTHCA** and **MPTHCA·2H_2_O**, despite the energies of the anhydrate form (−39.4 kJ mol^−1^) and hydrate ICC (−40.1 kJ mol^−1^) being almost comparable, formation of the hydrate ICC can lead to stronger interactions. However, it seems that estimations of lattice stabilization energy of ICCs may require deeper analysis.

## 4. Conclusions

Azole-containing compounds with varying biological activity have comprised 12.8% of approved small-molecule medicines, despite often having poor physicochemical and/or pharmacokinetic properties. Our CSD study reveals that multicomponent crystals based on the azolium-azole supramolecular heterosynthon are understudied. The synthesis and single crystal structures of 16 new ICCs based on imidazole and 1,2,4-triazole are presented herein to highlight the strong potential of azole groups to exhibit NH^+^···N supramolecular heterosynthons that can be exploited to generate ICCs. Whereas we attempted to isolate two kinds of ICCs, A^+^B^−^A and A^+^B^−^C, formation of ICCs based upon AH^+^···A supramolecular heterosynthons was more generally successful, consistent with CSD statistics of ICCs. Our CSD analysis showed that in imidazoles and 1,2,4-triazoles, the CNC angles are sensitive to protonation, and their cationic forms display larger values than those of the equivalent neutral rings, which is confirmed by the findings of this study. Three of the seventeen ICCs investigated herein formed isolated site hydrate ICCs, with two of them exhibiting both hydrate and anhydrate forms, both of which were sustained by charge-assisted NH^+^···N supramolecular heterosynthons. Analysis of calculated binding energies revealed that protonated azolium moieties offer a thermodynamically favourable NH^+^···N binding energy of 31.0 to 46 kJ mol^−1^. In this study mechanochemistry has proven to be a reliable and efficient method for synthesizing new ICCs using salts as a coformer. Application of this technique and others paves the way for the systematic construction of the robust and underexplored NH^+^···N supramolecular networks for the formation of ionic cocrystals of azole-based APIs to improve their physicochemical properties.

Overall, our results indicate that many parameters can be varied, including solvent, method, acids to enable isolation of azolium…azole ICCs. Furthermore, charge-assisted NH···N hydrogen bonding in A^+^B^−^A ICCs when the cation and neutral azole moiety are the same are not only statistically more likely, were calculated to offer more favourable energetic interactions. While the findings of this study are limited to ICCs of model coformers studied, they suggest that crystal engineering of azolium…azole ICCs for various purposes, including pharmaceutical products, is a viable crystal engineering strategy.

## Data Availability

The data presented in this study are available on request from the corresponding authors.

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
