# Peer review of "Crystal Engineering of Ionic Cocrystals Sustained by Azolium···Azole Heterosynthons"

_pharmaceutics, 2022, doi:10.3390/pharmaceutics14112321_

Round 1

Reviewer 1 Report

This manuscript describes the formation of an extensive series of ionic cocrystals (ICCs) containing various derivatives of imidazoles and triazoles. Overall, the work demonstrates the reliability of exploiting Azolium···Azole heterosynthons using various coformers in the presence of imidazole and triazole derivatives to form ICCs.

Different analyzes are combined to evaluate the reliability of this synthon, the SCXRD analysis being consistent with computational studies and using CSD. They also explored the preparation of these species using mechanochemistry and evaporative methods. It is mentioned that the mechanochemical methods were efficient for preparing the ICCs. Also, obtaining species comprising different azoles (AH+···C) was less successful.

In this way, I believe that the work presented both from the experimental point of view and computational calculations and CSD is well described.

However, I think the introduction is too long. I don't think what is described in lines 39-74 is necessary. Therefore, the authors are requested to fix the introduction and only consider what is described in lines 75-140.

Author Response

We thank the reviewer for reviewing our manuscript and his/her support of publication. The reviewer expresses concern about the length of the introduction.  Specifically,  “I don't think what is described in lines 39-74 is necessary. Therefore, the authors are requested to fix the introduction and only consider what is described in lines 75-140.”

Response:

We agree with the reviewer's comments about the length of the introduction and have modified accordingly. Specifically, the following portions of the introduction have been removed: historical aspects of crystal engineering; explaining the concept of supramolecular synthons and their classification; polymorphism. The importance of crystal engineering and cocrystals in pharmaceutical science is now addressed in the first paragraph.

Reviewer 2 Report

The Authors report a systematic crystal engineering study that evaluates the potential of azole compounds to reliably form  ionic cocrystals. They presented preparation, characterization, and structural studies of 16 new crystal forms of  ionic cocrystals comprising azole functional groups. They complemented their experimental studies with data mining from the Cambridge Structural Database and computational studies of intermolecular interaction energies of charge-assisted azolium-azole hydrogen bonds. Experimental design, synthesis and through characterization is very well taken care. This is a very valuable research on crystal engineering. The authors conclude that their results show that MCCs based on azole-azole supramolecular heterosyntons are a real target, which has an impact on the development of new azole drugs with improved properties. Unfortunately they do not provide any example of ionic cocrystals with API  comprising azole functional groups, particularly imidazoles and triazoles with improved properties. They could show (on miconazole or other API, for example) that the miconazole ionic co-crystals showed higher dissolution rates than the free base. The article is great, but I wonder if the journal is appropriate. I think Crystal Growth & Design or  CrystEngComm would be better. If it is to be published in Pharmaceuticals, references must be corrected and completed (e.g. 32, 59,…).

Author Response

We thank the reviewer for reviewing our manuscript and his/her support of publication.

With respect to reviewer's request to demonstrate that our approach works for drug molecules such as miconazoles, this is indeed a subject that is being pursued by us in order to control physicochemical properties. However, the purpose of the current manuscript is to establish crystal engineering principles on azole model compounds in general, an approach we have taken previously with articles published in journals such as Molecular Pharmaceutics and Journal of Pharmaceutical Science.

In short, we understand the reviewer’s concern about the relevance of our manuscript to Pharmaceutics, but it is our modus operandi to separate crystal engineering studies from pharmaceutical science studies. Further, the theme of this special issue of Pharmaceutics is crystal engineering and its relevance to pharmaceutical science. We intend to publish a separate paper on miconazole based pharmaceutical cocrystals and their physicochemical properties shortly after this article is published.

Reviewer 3 Report

 Rahmani et al reported the synthesis and structural characterizations of 16 new ionic cocrystals based on imidazole and 1,2,4-triazole to highlight the NH+···N supramolecular hetero-synthons which is responsible for cocrystals. In my opinion, the manuscript is well written, and all new solids are also characterized in detail. I think the paper can be accepted for publication after the author considers the following:

1.       The PXRD patterns of simulated and experimental of synthesized cocrystals are different. The author should provide the reason for that.

2.       Include the following reference on cocrystallization for improving the property.

Cryst. Growth Des. 2019, 19, 11, 6482–6492

Author Response

We thank the reviewer for reviewing our manuscript and his/her support of publication.  We have addressed the specific requests of the reviewer as follows:

  1. The PXRD patterns of simulated and experimental of synthesized cocrystals are different. The author should provide the reason for that.

Response: We agree that some of our plots required replotting on a more appropriate y scale and we have done so. We also recollected PXRD data in several instances. There is always likely to be slight variation in unit cell parameters as a function of temperature. In this context, the agreement between the experimental PXRD patterns determined at room temperature and those calculated using the SCXRD date (at low temperatures) is good as shown in Figs S3.1-S3.15.

  1. Include the following reference on cocrystallization for improving the property Cryst. Growth Des. 2019, 19, 11, 6482–6492

Response: This study is now cited as ref 16.

Round 2

Reviewer 2 Report

It is undoubtedly a very valuable publication. Perhaps extending to include information linking the crystal engineering of azoles with their physicochemical properties would make it too long. So I am waiting for the publication promised by the Authors on miconazole based pharmaceutical cocrystals and their physicochemical properties.